# Colicin N Mediates Apoptosis and Suppresses Integrin-Modulated Survival in Human Lung Cancer Cells

**DOI:** 10.3390/molecules25040816

**Published:** 2020-02-13

**Authors:** Wanatchaporn Arunmanee, Gea Abigail U. Ecoy, Hnin Ei Ei Khine, Methawee Duangkaew, Eakachai Prompetchara, Pithi Chanvorachote, Chatchai Chaotham

**Affiliations:** 1Department of Biochemistry and Microbiology, Faculty of Pharmaceutical Sciences, Chulalongkorn University, Bangkok 10330, Thailand; wanatchaporn.a@chula.ac.th (W.A.); ecoygea@gmail.com (G.A.U.E.); hnineieikhine12@gmail.com (H.E.E.K.); d.methawee@gmail.com (M.D.); 2Vaccines and Therapeutic Proteins Research Group, the Special Task Force for Activating Research (STAR), Faculty of Pharmaceutical Sciences, Chulalongkorn University, Bangkok 10330, Thailand; eakachai.p@chula.ac.th; 3Department of Pharmacy, School of Health Care Professions, University of San Carlos, Cebu 6000, Philippines; 4Department of Laboratory Medicine, Faculty of Medicine, Chulalongkorn University, Bangkok 10330, Thailand; 5Center of Excellence in Vaccine Research and Development (Chula Vaccine Research Center-Chula VRC), Faculty of Medicine, Chulalongkorn University, Bangkok 10330, Thailand; 6Department of Pharmacology and Physiology, Faculty of Pharmaceutical Sciences, Chulalongkorn University, Bangkok 10330, Thailand; 7Cell-based Drug and Health Products Development Research Unit, Faculty of Pharmaceutical Sciences, Chulalongkorn University, Bangkok 10330, Thailand

**Keywords:** Colicins, Apoptosis, Integrin, selective anticancer, human lung cancer cells

## Abstract

The inherent limitations, including serious side-effects and drug resistance, of current chemotherapies necessitate the search for alternative treatments especially for lung cancer. Herein, the anticancer activity of colicin N, bacteria-produced antibiotic peptide, was investigated in various human lung cancer cells. After 24 h of treatment, colicin N at 5–15 µM selectively caused cytotoxicity detected by MTT assay in human lung cancer H460, H292 and H23 cells with no noticeable cell death in human dermal papilla DPCs cells. Flow cytometry analysis of annexin V-FITC/propidium iodide indicated that colicin N primarily induced apoptosis in human lung cancer cells. The activation of extrinsic apoptosis evidenced with the reduction of c-FLIP and caspase-8, as well as the modulation of intrinsic apoptosis signaling proteins including Bax and Mcl-1 were observed via Western blot analysis in lung cancer cells cultured with colicin N (10–15 µM) for 12 h. Moreover, 5–15 µM of colicin N down-regulated the expression of activated Akt (p-Akt) and its upstream survival molecules, integrin β1 and αV in human lung cancer cells. Taken together, colicin N exhibits selective anticancer activity associated with suppression of integrin-modulated survival which potentiate the development of a novel therapy with high safety profile for treatment of human lung cancer.

## 1. Introduction

Cancer is a global health burden and non-small cell lung carcinoma (NSCLC) is the primary cause of cancer-related deaths worldwide [1]. Clinical and in vitro evidence indicate that lung cancer cells frequently manifest multi-drug resistance to apoptosis via standard chemotherapy, leading to patient mortality [2,3,4]. Despite several decades of research, the 5-year survival rate of lung cancer patients has not been significantly improved [2]. Thus, the search for novel anticancer drugs providing better patient outcomes is urgently needed.

To seek novel anticancer drugs, the current paradigm holds that anticancer agents induce toxicity in cancer cells through restoration of normal apoptosis signals [5,6,7]. Apoptosis is a type of programmed cell death that is vital for cell homeostasis and can be initiated through extrinsic and intrinsic signaling pathways in response to various stimuli [6,8]. To restore normal apoptosis signals, in vivo and in vitro studies show that suppression of anti-apoptosis Bcl-2 (B-cell lymphoma 2) and Mcl-1 (Myeloid cell leukemia 1) protein as well as up-regulation of pro-apoptosis protein, Bax (Bcl-2-associated X protein) strongly correlates with apoptosis induction in cancer cells, including those obtained from advanced grade lung cancer specimens [9,10]. Clearly, the disrupted balance between anti- and pro-apoptosis Bcl-2 family proteins in the intrinsic apoptosis pathway is involved in carcinogenesis and resistance to cell death [4]. Moreover, cancer cells also evade death receptor-mediated extrinsic apoptosis through the overexpression of c-FLIP (cellular FLICE-like inhibitory protein), a master anti-apoptosis protein. The inhibitory effect of c-FLIP on caspase-8 and caspase-10 leads to the suppression of both extrinsic and intrinsic apoptosis pathways [11,12]. The augmented level of c-FLIP, Bcl-2 and Mcl-1 has been noted in NSCLC, posing present and future challenges in cancer therapy [11,13,14,15].

Furthermore, suppression of the integrin/Akt (Protein kinase B) survival signal has been recognized as a novel strategy to induce apoptosis and restore drug sensitivity in lung cancer cells [16,17,18]. The ligation of integrins on the cell membrane to extracellular matrix (ECM) initiates pro-survival PI3K (Phosphatidylinositol-4,5-bisphosphate 3-kinase)/Akt signal. Activated Akt (phosphorylated-Akt; p-Akt) subsequently mediates the expression of anti-apoptosis proteins including Bcl-2, Mcl-1 and c-FLIP [16,19]. Lung cancer cells have been found to up-regulate integrin transmembrane proteins, enabling increased cell adhesion and PI3K/Akt survival activity [20]. A recent trend in chemotherapy is the emergence of β1 and αV integrins as therapeutic targets due to their cooperative role in regulation of survival, metastasis and drug resistance especially noted in NSCLC [16,20,21]. The overexpression of integrin β1 has been shown to be a poor prognostic factor in NSCLC associated with drug resistance [20,22]. β1 integrin is particularly important in lung cancer, as several studies suggest, and one of its prominent binding partners is αV integrin, which has also been noted to be up-regulated in lung cancer [23]. The overexpression of αVβ1 has also been implicated in the resistance of lung cancer cells against radiation and various chemotherapeutic agents [22]. Taken together, the modulation of integrin β1 and αV is a worthy strategy in the search for novel and improved therapeutic options for lung cancer treatment.

Unprecedented biotechnology advancements, particularly in bioprocessing, has allowed the easy production of proteins and peptides to become more feasible and practical therapeutic options [24,25,26]. Most importantly, natural peptides/proteins have gained recognition as novel anticancer agents [25,27,28]. Bacteria-produced antimicrobial peptides/proteins such as colicins are deemed to have further application in cancer therapy because of reported selective toxicity in various cancer cells [25,29,30]. Colicins are proteins naturally produced by *Escherichia coli* (*E. coli*) in stress conditions to potently inhibit the growth of other related bacteria. The mode of bactericidal action is often used to classify the colicins: pore-formers, nuclease types and anti-murein synthesis colicins [31]. RNase colicins and pore-forming colicins exert cytotoxicity on tumor cells, with the pore-formers showing the strongest effect and thus, good potential for investigation [28]. Although accumulating research in colicins is warranted, as in vitro studies have revealed apoptosis induction in several cancer cells and in vivo mice studies have shown reduced tumor burden after colicin treatment, the certain anticancer mechanism of colicins is not well established [25,28]. Furthermore, therapeutic application of colicin N, the smallest (42 kDa) representative of the pore-forming colicins, for NSCLC treatment has not been investigated. To validate the potential development of an effective anticancer peptide, the cytotoxicity and apoptosis-inducing effects as well as related anticancer mechanisms of colicin N in human non-small cell lung cancer cells was elucidated in this study.

## 2. Results

### 2.1. Purified Colicin N Expressed in Recombinant E. coli

Colicin N comprising all domains was successfully expressed in *E. coli* BL21-AI from a plasmid encoding a C-terminal Histidine-tagged Colicin N gene in a pET3a vector. Histidine-tagged colicin N was then purified by using a nickel-sepharose HisTrap™ HP affinity column, where it was strongly retained. Unbound proteins were washed with wash buffer and represents the first peak in the elution profile of colicin N (Figure 1a). The addition of the elution buffer with increased concentration of the competitive ligand, imidazole, corresponded to increased gradient concentration and a sharp peak of eluted fraction (EF) in the chromatogram. The purity of protein confirmed by SDS-PAGE showed that most contaminants were removed and the expected band of colicin N at 40 kDa was observed (Figure 1b). Additionally, the bactericidal activity against *E. coli* tested by broth microdilution method was demonstrated to evaluate a biological function of the expressed colicin N (data not shown).

### 2.2. Colicin N Causes Toxicity in Human Lung Cancer Cells

Preliminary evaluation of cytotoxicity against NSCLC was performed in human lung cancer H460 cells maintained in culture medium containing 0–15 µM colicin N for 24 h. MTT assay showed the significant reduction of %cell viability in the cells exposed with colicin N at 1–15 µM compared with non-treated control cells (Figure 2a). Consistent with the viability results, colicin N-induced cell death was also observed. Co-staining of Hoechst33342/propidium iodide (PI) revealed the effect of colicin N on induction of apoptosis in human lung cancer cells (Figure 2b). Hallmark features of apoptosis such as DNA condensation and nuclei fragmentation were observed with the bright blue fluorescence of Hoechst33342 staining in H460 cells incubated with 5–15 µM of colicin N in a concentration-dependent manner (Figure 2c). Notably, the observation of colicin N-treated H460 cells under fluorescence microscopy detected no red fluorescent cells permeated with PI staining, which is characteristic of necrosis cells with compromised membrane integrity.

### 2.3. Colicin N-induced Apoptosis in Human Lung Cancer Cells

Mode of cell death in colicin N-treated human lung cancer cells was further confirmed by flow cytometry analysis of annexin V-FITC/PI staining. Translocation of phosphatidylserine to outer leaflet of cell membrane is a key event that occurs prior to end-stage DNA fragmentation in apoptosis process [8]. Therefore, the specific binding of annexin V-FITC to phosphatidylserine sensitively detects apoptosis at early stage [32]. Consistent with the cell death detected by co-staining of Hoechst33342/PI, flow cytometry revealed that colicin N at 5–15 µM increased apoptosis at both early (annexin V-FITC^+^/PI^−^) and late (annexin V-FITC^+^/PI^+^) stages in a concentration-dependent manner (Figure 3a). Interestingly, higher %apoptosis was noted in annexin V-FITC/PI flow cytometry analysis (Figure 3b) compared with nuclear staining with Hoechst33342/PI (Figure 2b). In summary, these results demonstrate apoptosis-inducing effect of colicin N in human lung cancer cells.

### 2.4. Colicin N Activates both Intrinsic and Extrinsic Apoptosis Pathways

To clarify underlying mechanisms of colicin N on induction of apoptosis, the alteration of apoptosis-regulating proteins was examined by Western blot analysis. Figure 4a,b provides evidence of the activation of extrinsic apoptosis pathway in human lung cancer cells cultured with 5–15 µM colicin N for 12 h. The reduction of c-FLIP, a known inhibitor of caspase-8 was observed, along with the decrease of caspase-8 and the increase of cleaved caspase-8 level. Such evidence suggests that death receptor-mediated apoptosis might be activated in colicin N-treated H460 cells. Another important facet investigated was the regulatory effect of colicin N on Bcl-2 family proteins that control the intrinsic/mitochondrial apoptosis. The level of anti-apoptosis protein, Mcl-1, obviously decreased in H460 cells after incubation with 10–15 µM colicin N while there was no alteration of Bcl-2 protein level (Figure 4c). Mitochondrial permeabilization during apoptosis is dependently on Bax oligomerization which can be regulated by Mcl-1 [33]. Correlation with reduction of Mcl-1, Figure 4d indicates the up-regulation of Bax in H460 cells exposed with 10-15 µM colicin N for 12 h. The obtained results imply that colicin N mediates intrinsic apoptosis via modulation of Mcl-1 and Bax expression level.

### 2.5. Inhibitory Effect of Colicin N on Integrin/Akt Survival Signal in Human Lung Cancer Cells

Upstream signals known to regulate apoptosis were further investigated in human lung cancer cells treated with colicin N. Integrin-mediated pro-survival pathways are studied intensely in the cancer field for novel targeted therapy [22,34], and were investigated in this study on colicin N activity. Figure 4e reveals the attenuation of integrin β1 and αV in human lung cancer H460 cells treated with 15 µM of colicin N for 12 h. Interestingly, significant reduction of integrin β1 was found even with lower concentrations (5–10 µM) of colicin N. In connection, alteration of Akt and Erk, which are downstream survival molecules of integrins was demonstrated in Figure 4f. Down-regulation of activated Akt (p-Akt) was significantly presented in colicin N-treated H460 cells, but no significant alteration of the level of total Akt was noted. Moreover, colicin N at 5–15 µM did not remarkably change the expression of Erk and p-Erk in human lung cancer cells. Considering this, the decrease in cell viability and survival in colicin N treated-lung cancer cells might be primarily mediated by integrin/Akt signaling pathway.

### 2.6. Selective Apoptosis-Inducing Effect of Colicin N in Various Human Lung Cancer Cells

To confirm anticancer activity in various lung cancer cells, cytotoxicity and mode of cell death were evaluated in human lung cancer H292 and H23 cells cultured with 0–15 µM colicin N for 24 h. Figure 5a,c, respectively, demonstrates significant reduction of %cell viability in H292 and H23 cells after treatment with colicin N at 5–15 µM. Although toxicity of colicin N seems to lower in p53 mutant H23 cells when compared with lung cancer cells with p53 wild type H460 and H292 cells as evidenced by the cell viability assay, nuclear staining clearly reveals DNA condensation/nuclei fragmentation presenting apoptosis in H292 and H23 cells (Figure 5b,d, respectively).

Hair loss is one of major side effects that lessen quality of life in lung cancer patients administered with current standard chemotherapy [35,36,37]. Apoptosis in dermal papilla cells caused by anticancer drugs leads to chemotherapy-induced alopecia [38,39,40]. Therefore, the toxic profile of colicin N in normal cells was investigated in human dermal papilla DPCs cells. Approximately 10% reduction of cell viability was detected through MTT assay in DPCs cells incubated with colicin N at the highest concentration, 15 µM for 24 h (Figure 5e). Nevertheless, neither apoptosis nor necrosis was presented in DPCs cells after colicin N treatment (Figure 5f). These results confirm that colicin N selectively induced cell death in human lung cancer cells.

## 3. Discussion

Decreased sensitivity to cell death is a key factor for innate and acquired drug resistance to classical therapy in lung cancer [4,20,41,42]. The up-regulation of survival signals and anti-apoptosis proteins regulating both intrinsic (Bcl-2 and Mcl-1) and extrinsic (c-FLIP) pathway is a phenomenon noted in lung cancer specimens and cell lines, causing high resistance to chemotherapy induced-cell death [11,13,15,20]. Therefore, targeting Akt survival pathway and apoptosis machinery is particularly relevant in lung cancer treatment. The data herein present apoptosis-inducing effect of colicin N, a bacteriocin, in human lung cancer cells. Not only diminution of Mcl-1 and c-FLIP but also up-regulation of Bax, a pro-apoptosis protein, was presented in human lung cancer cells exposed to colicin N (Figure 4a–d). This alteration of apoptosis-regulating proteins correlates with dramatically increased apoptosis in colicin N treated-lung cancer cells (Figure 2 and Figure 3). Mcl-1 is the primary anti-apoptosis protein overexpressed in various NSCLC, implying that Mcl-1 inhibition could lead to more successful targeted therapy for lung cancer [15]. Moreover, down-regulation of Mcl-1 and c-FLIP is therapeutically relevant. Up-regulation of Mcl-1 has been linked to advanced stage of tumor pathology and metastasis while c-FLIP overexpression significantly correlates to shorter survival in NSCLC patients [42,43].

Akt has been recognized as an upstream molecule up-regulating anti-apoptosis proteins. The transcription of Bcl-2 can be activated by Akt while sustained levels of Mcl-1 and c-FLIP come from the inhibitory effect of Akt on protein degradation [11,44]. A promising anticancer activity is the weakening of Akt survival signal as indicated by the reduction of activated Akt (p-Akt) in human lung cancer cells cultured with colicin N (Figure 4e,f). Despite no significant alteration of Bcl-2 level, the reduction of Mcl-1 and c-FLIP observed upon colicin N treatment correlates with the suppression on p-Akt signaling. In line with its multi-functional role, c-FLIP also interacts with Akt to enhance the anti-apoptosis functions of Akt [12]. It should be noted that colicin N might mainly modulate apoptosis-regulating proteins through post-translational modification. Akt is also known to indirectly restrain the activity of Bax through inhibiting Bax translocation to mitochondrial membrane [10,33,45]. The up-regulation of Bax has been found to be directly proportional to the level of apoptosis observed in locally advanced lung cancer tissue [9]. Although the mechanism of Akt up-regulating Bax is still needs further clarification, various natural compounds modulating Akt survival pathway have been found to simultaneously alter Bax expression [46,47]. Likewise, suppression of Akt associated with the up-regulated Bax was demonstrated in colicin N-treated human lung cancer cells (Figure 4).

The relationship between aggressive features including chemo-resistance and expression level of the transmembrane proteins, integrins, is well established in NSCLC [20]. Integrin-ECM interaction generates outside-in signaling consequently activating PI3K/Akt survival pathway [16,20]. The reduction of p-Akt level might result from down-regulation of integrin β1 and αV in lung cancer cells after treatment with colicin N (Figure 4e,f). Suppression of integrin-mediated survival has been highlighted in the development of anticancer peptides [48]. Antagonistic effect on specific integrin motifs has been proposed to disrupt integrin-ECM interaction following with internalization and eventually degradation of integrins [23,49]. Despite the observed round morphology of colicin N treated-human lung cancer cells (data not shown), the inhibitory effect of colicin N on the interaction between specific integrins and ECM components consequence with the alteration of downstream signaling molecules need to be further investigated.

The cooperative network of integrins and downstream signaling molecules not only mediates cell survival and death but also activates metastatic behavior [50]. Among many sub-types of integrins, β1 integrin in coordination with Akt is notoriously exploited by NSCLC to acquire resistance to cell death induced by anticancer drugs [20,50]. Overexpression of integrin αV correlates with metastasis lesion in various cancer cells including lung cancer [51]. It is worth noting that the modulation on other sub-types of integrin and extended benefits on metastasis prevention and chemo-sensitization of colicin N are interested in future studies.

Serious side effects in normal cells mainly detract from therapeutic benefits of effective anticancer compounds [52]. Due to its high proliferative rate being similar to cancer cells, the hair follicle is an organ negatively affected by various chemotherapies [53]. Most of the recommended anticancer drugs for lung cancer significantly induced apoptosis in dermal papilla cells consequently causing drug induced-alopecia and low quality of life in cancer patients [38,39,40]. The absence of apoptosis in human dermal papilla DPCs cells treated with colicin N strongly supports the development of colicin N as a selective anticancer drug for treatment of lung cancer. Colicin N is a cationic antimicrobial pore-forming protein, and like most colicins, its antimicrobial mechanism has been thoroughly investigated. Interestingly, it has been suggested that pore-forming colicins induce cytotoxicity and apoptosis by generating pores in the plasma membrane of cancer cells [25,28]. Bacteriocins such as colicins are deemed to exert selectivity towards cancer cells due to surface factors that differentiate cancer cells from normal cells [25]. Consideration of negative charge and overexpression of specific integrins on cell membrane of lung cancer cells is a point of interest that should be further investigated for selective anticancer activity of colicin N.

## 4. Materials and Methods

### 4.1. Chemical Reagents

For protein expression, arabinose was obtained from Merck (Billerica, MA, USA). Monobasic sodium phosphate was purchased from Vivantis Technologies (Selangor Darul Ehsan, MY, USA). Imidazole for buffer solutions, DNase I and ampicillin sodium salt were bought from PanReac Applichem (Darmstadt, DE, USA). Luria-Bertani (LB) broth and agar were acquired from Hardy Diagnostics (Santa Maria, CA, USA). AccuPrep^®®^ Nano-Plus plasmid mini extraction kit was sourced from Bioneer Inc. (Alameda, CA, USA). The protease inhibitor tablets and PierceTM bicinchoninic acid (BCA) protein assay kit was procured from Thermo Fisher Scientific (Waltham, MA, USA).

Roswell Park Memorial Institute (RPMI) 1640 medium, phosphate-buffered saline (PBS); pH 7.4, trypsin, l -glutamine, fetal bovine serum (FBS) and penicillin/streptomycin solution were obtained from Gibco (Gaithersburg, MA, USA). Apoptosis measurement kit with Annexin V-Fluorescein Isothiocyanate (FITC) was procured from Thermo Fisher Scientific (Waltham, MA, USA). Hoechst33342, propidium iodide (PI), 3-(4,5-Dimethylthiazol-2-yl)-2,5-diphenyltetrazolium bromide (MTT), dimethysulfoxide (DMSO), TRIS hydrochloride (Tris–HCl), sodium chloride (NaCl), Tween 20, skim milk, bovine serum albumin (BSA) and ammonium sulfate were purchased from Sigma Chemical, Inc. (St. Louis, MO, USA). The protease inhibitor cocktail was procured through Roche Molecular Biochemicals (Indianapolis, IN, USA). Cell Signaling Technology, Inc. (Denver, MA, USA) was a source of antibodies including primary antibody of Bcl-2, Mcl-1, Bax, caspase-8, c-FLIP, Akt, phosphorylated-Akt (p-Akt; Ser 473), extracellular signal–regulated kinase (Erk), phosphorylated-Erk (p-Erk; Thr 981), integrin β1, integrin αV and β-actin as well as peroxidase-labeled secondary antibodies. Immobilon Western chemiluminescent horse radish peroxidase substrate was bought from Millipore, Corp (Billerica, MA, USA).

### 4.2. Expression of Colicin N

pET3a plasmids encoding c-terminal histidine-tagged colicin N were transformed into BL21-AI™ One Shot^®®^ chemically competent *E. coli* (Invitrogen, Thermo Fisher Scientific, Waltham, MA, USA). The transformants were selected using LB agar plate containing 100 μg/mL ampicillin maintained at 37 °C overnight. The selected bacteria were grown into LB media up to Optical Density (OD) 600 of 0.6, then colicin N expression was induced by the addition of arabinose at the final concentration of 0.2% (*w*/*v*). Three hours after induction, the cells were harvested by centrifugation for 10 min at 8000× *g* at 4 °C then resuspended in a binding buffer (50 mM sodium phosphate buffer; pH 8.0, 300 mM NaCl and 10 mM imidazole). RNase, DNase I, and protease inhibitors were added to this suspension and cell lysis was subsequently performed by pulse sonication for 15 min on ice. Finally, the supernatant was obtained by centrifugation at 12,000× *g* at 4 °C for 10 min.

### 4.3. Purification of Colicin N

Colicin N was purified from the supernatant by affinity chromatography using a nickel-sepharose HisTrap™ HP affinity column (GE Healthcare Technologies, West Milwaukee, WI, USA) in a Fast Protein Liquid Chromatography (FPLC) ÄKTA Start machine (GE Healthcare Technologies, West Milwaukee, WI, USA). The column was pre-equilibrated with the binding buffer and the supernatant was applied via sample pump. The column was washed with the washing buffer (50 mM sodium phosphate buffer; pH 8.0, 300 mM NaCl and 20 mM imidazole) then colicin N was eluted with a buffer containing 50 mM sodium phosphate buffer (pH 8.0), 300 NaCl and 250 mM imidazole. The protein-containing fractions were lyophilized and resuspended in PBS. The protein concentration was measured by Pierce™ BCA kit (Thermo Fisher Scientific, Waltham, MA, USA). The purity colicin N were analyzed by sodium dodecyl sulfate polyacrylamide gel electrophoresis (SDS-PAGE). Moreover, antibacterial activity of expressed colicin N was confirmed by broth microdilution assay in microplate format, with LB media for the growth of *E. coli* 8739™ (ATCC, Manassas, VA, USA).

### 4.4. Cell Culture

Human lung cancer H460, H292 and H23 cells (ATCC, Manassas, VA, USA) were maintained in RPMI 1640 medium supplemented with 2 mM l-glutamine, 10% FBS and 100 units/mL of penicillin/streptomycin. Meanwhile, human dermal pupilar DPCs cells obtained from Applied Biological Materials Inc. (Richmond, Canada) were cultured Prigrow III medium (Applied Biological Materials Inc., Richmond, BC, Canada) containing l-glutamine (2 mM), FBS (10%) and of penicillin/streptomycin (100 units/mL). All cells were incubated in appropriate condition of 5% CO_2_ at 37 °C until 70%–80% confluence before used in further experiments.

### 4.5. Cytotoxicity Assay

Cells seeded in 96-well plate at a density of 1 × 10^4^ cells/well were incubated with various concentrations (0–15 µM) of colicin N for 24 h. Then, MTT assay was performed to evaluate cell viability. After removal of culture medium, the cells were further incubated with MTT (0.4 mg/mL) and kept from light at 37 °C for 4 h. Then MTT solution was replaced with DMSO to dissolve the formazan crystal product. The intensity of the formazan color was measured via microplate reader (Anthros, Durham, NC, USA) at 570 nm. The ratio of optical density (OD) between treated cells to non-treated control cells was represented as percent cell viability.

### 4.6. Detection of Mode of Cell Death

Mode of cell death was initially inspected through Hoechst33342 and PI nuclear co-staining. After 24 h of treatment with colicin N, cells were incubated for 30 min with PBS (pH 7.4) containing 10 µM Hoechst33342 and 5 µg/mL PI dyes at 37 °C. Then, the cells were observed under a fluorescence microscope (Olympus IX51 with DP70, Olympus, Tokyo, Japan). Apoptosis cells with condensed chromatin and/or fragmented nuclei are presented by bright blue fluorescence of Hoechst33342. Meanwhile, PI-stained cells emit red fluorescence and represent necrosis cell death.

### 4.7. Flow Cytometry Analysis of Annexin V-FITC/PI

Flow cytometry using an annexin V-FITC apoptosis assay kit was additionally executed to evaluate and quantify apoptosis and necrosis cell death. Colicin N-treated lung cancer cells were harvested and prepared into single-cell suspension in PBS (pH 7.4). After centrifugation at 5000 rpm × 5 min at 4 °C, the cell pellets were resuspended in 100 µL of binding buffer. Annexin V-FITC (1 μg/mL) and PI (2.5 μg/mL) solution were added into the cell suspensions as recommended in the manufacturer’s instructions. Cells in the samples are classified as living, apoptosis and necrosis cells via a Guava^®®^ easyCyte 5 benchtop flow cytometer and the analysis was performed with guavaSoft ™ version 2.7 software (Merck, Darmstadt, Germany, 2013).

### 4.8. Western Blot Analysis

After human lung cancer cells (5 × 10^5^ cells/well in a 6-well plate) were maintained in complete medium containing colicin N at 0–15 µM for 12 h, the cells were washed with cold PBS (pH 7.4) and further incubated on ice for 45 min with a lysis buffer containing 20 mM Tris-HCl (pH 7.5), 1% Triton X-100, 150 mM sodium chloride, 10% glycerol, 1 mM sodium orthovanadate, 50 mM sodium fluoride, 100 mM phenylmethylsulfonyl fluoride, and a protease inhibitor cocktail. The supernatant from the cell lysate was collected after centrifugation at 8000 rpm at 4 °C for 15 min. Samples were prepared to contain equal amount of protein content which was determined through BCA protein assay. Each sample was denatured with Laemmli loading buffer at 95 °C for 5 min. After resolving through 10% SDS-PAGE, the separated proteins were transferred onto 0.45 μm nitrocellulose membrane (Bio-Rad, Hercules, CA, USA). The membrane was incubated with non-fat dry milk (5% in TBST; 25 mM Tris-HCl; pH 7.5, 125 mM NaCl and 0.05% Tween 20) for 1 h at 25 °C to prevent the detection of non-specific proteins. The membrane was washed three times (7 min) with TBST then the protein of interest was detected by incubation with specific primary antibody at 4 °C for 12 h. After washing with TBST (3 times × 7 min), horseradish peroxidase-labeled isotype-specific secondary antibody was added onto the membrane for 2 h at room temperature. Before detection of the immune complex, the membrane was washed again with TBST (3 times × 7 min) then the signal from adding chemiluminescence substrate on the membrane was quantified by analyst/PC densitometric software (Bio-Rad Laboratory, Hercules, CA, USA, version 6.0.1, 2017).

### 4.9. Statistical Analysis

Mean data were averaged from three independent experiments. Statistical analysis was performed on SPSS Statistic 22 version using one-way ANOVA with Tukey post hoc test. A *p*-value ≤ 0.05 was considered as statistically significant.

## 5. Conclusions

The novel data obtained in this study demonstrate the apoptosis-inducing effect of colicin N, a bacteriocin protein, in human lung cancer cells through the modulation on integrin/Akt survival pathway associated with activation of both intrinsic and extrinsic apoptosis cascade (Figure 6). The down-regulation of αV and β1 integrins, which are frequently overexpressed in lung cancer, may also provide the additional benefit of circumventing drug resistance often encountered in current therapy. Additionally, the selective toxicity of colicin N as evidenced by its low toxicity towards actively dividing human dermal papilla cells indicate its advantage over conventional chemotherapy in terms of safety profile. Thus, these results highlight the potential anticancer activity of colicin N, warranting further development as an effective chemotherapy.

## Figures and Tables

**Figure 1 molecules-25-00816-f001:**
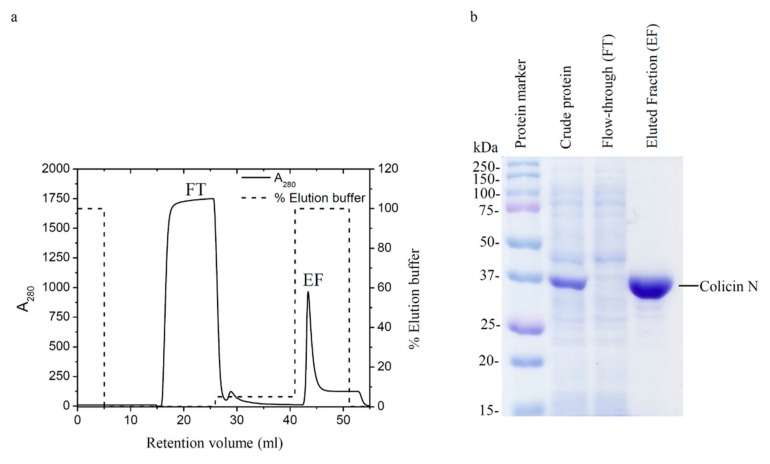
Purification of colicin N (**a**) Elution profile (black line) of colicin N using a nickel-sepharose HisTrap™ HP affinity column pre-equilibrated with a binding buffer (50 mM sodium phosphate buffer; pH 8.0, 300 mM NaCl and 10 mM imidazole). A 100% of elution buffer (50 mM sodium phosphate buffer, pH 8.0, 300 mM NaCl and 250 mM imidazole) was applied to the column for eluting colicin N. The percentage of elution buffer is shown as a dash line. (**b**) SDS-PAGE of crude protein and protein-containing fractions taken from the column. The band corresponding to colicin N shows at ~40 kDa.

**Figure 2 molecules-25-00816-f002:**
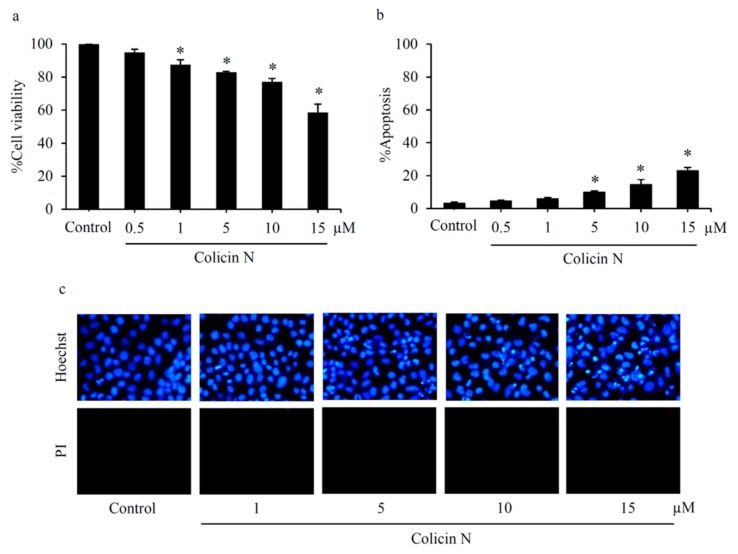
Apoptosis-inducing effect of colicin N in human lung cancer cells (**a**) Reduction in cell viability detected by MTT assay of lung cancer H460 cells was observed after treatment with colicin N (1–15 µM) for 24 h. (**b**) Significantly concentration-dependent increase in apoptosis occurred after colicin N treatment. (**c**) Co-staining with Hoechst33342 and propidium iodide (PI) reveals blue fluorescence of apoptosis in H460 cells treated with 5–15 µM of colicin N for 24 h. Meanwhile, there was no noticeable necrosis presenting with red fluorescence. Values are means of the independent triplicate experiments ± SD. * *p* ≤ 0.05 versus non-treated control.

**Figure 3 molecules-25-00816-f003:**
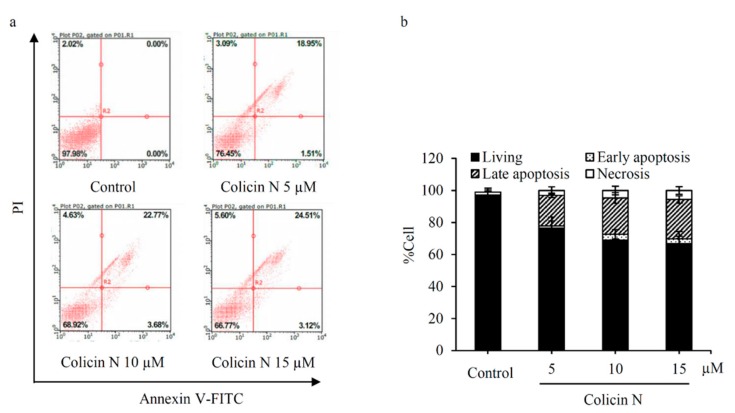
Flow cytometry analysis of mode of cell death in colicin N-treated lung cancer cells (**a**) Annexin V-FITC/propidium iodide (PI) histograms were obtained from analysis of lung cancer H460 cells incubated with 5–15 µM of colicin N for 24 h. Treatment with colicin N significantly increased apoptosis indicating with annexin V-FITC positive cells. (**b**) The decrease of %living cells corresponding with the augmentation of early and late apoptosis was observed in colicin N-treated H460 cells. Notably, flow cytometry showed minor alteration of necrosis cells after treatment the cells with colicin N. Representative experiment out of three is shown.

**Figure 4 molecules-25-00816-f004:**
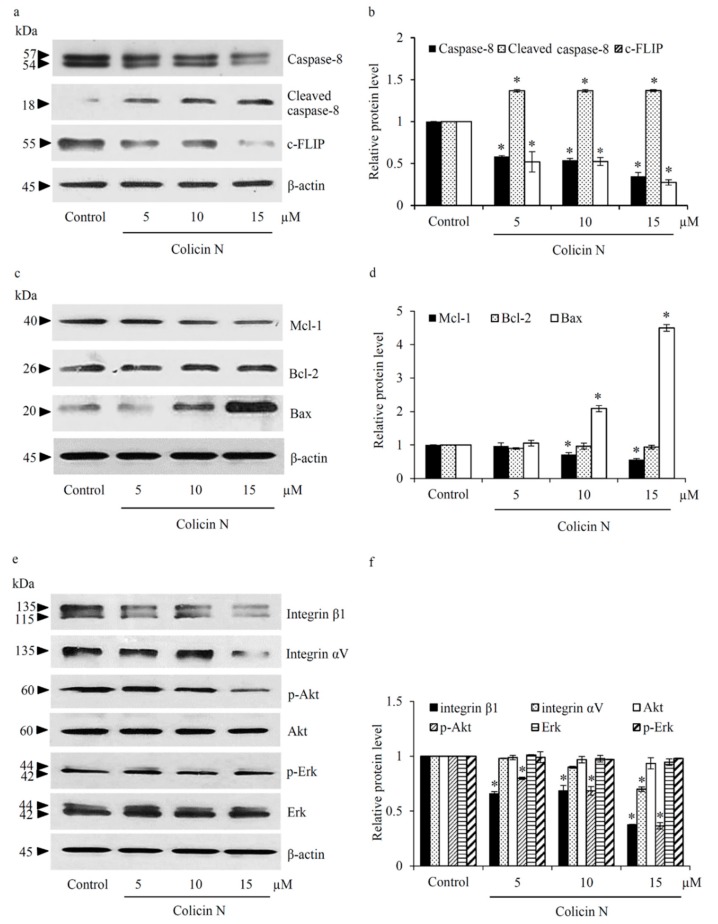
Colicin N suppresses integrin mediated-survival pathways. (**a**) Western blot analysis revealed the alteration of regulatory proteins related to extrinsic apoptosis including caspase-8 and c-FLIP. (**b**) The increase of cleaved caspase-8 accompanied by down-regulation of caspase-8 and c-FLIP was observed after culture lung cancer H460 cells with colicin N at 5–15 µM for 12 h. (**c**) Colicin N mediated mitochondrial apoptosis signal through up-regulation of pro-apoptosis protein, Bax and decreased anti-apoptosis Mcl-1 protein. (**d**) Meanwhile, there was no significant alteration of Bcl-2 level in colicin N-treated H460 cells. (**e**) Down-regulation of integrin β1 and αV were demonstrated in lung cancer H460 cells cultured with 15 µM of colicin N for 12 h. Interestingly, the reduction of integrin β1 was also noted in the cells treated with 5-10 µM colicin N. (**f**) Treatment with colicin N (5–15 µM) obviously suppressed p-Akt level while Erk and its activated form (p-Erk) was fairly altered in H460 cells. Values are means of the independent triplicate experiments ± SD. * *p* ≤ 0.05 versus non-treated control.

**Figure 5 molecules-25-00816-f005:**
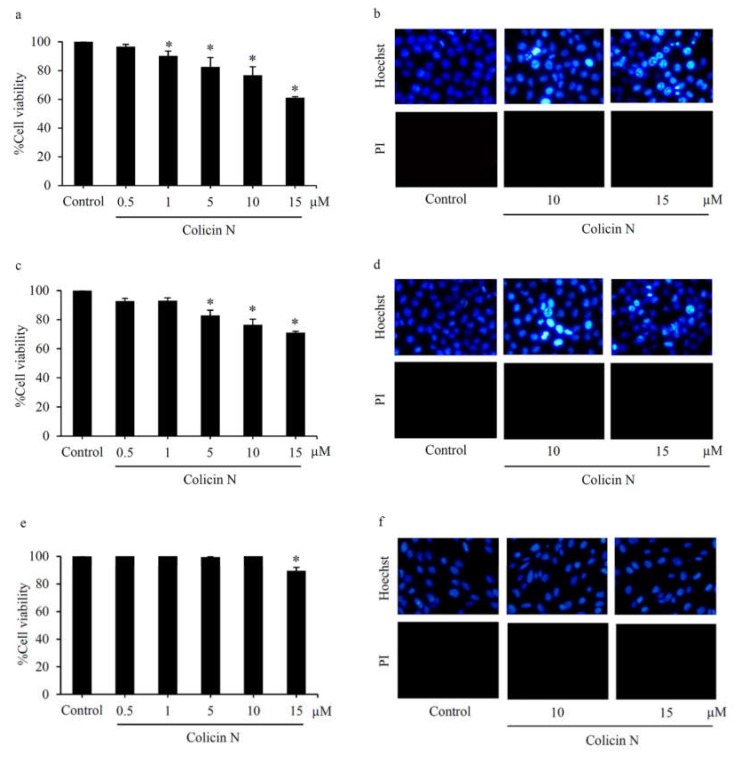
MTT viability assay of (**a**) H292 and (**c**) H23 cells, depicting anticancer activity of colicin N (5–15 µM) against various types of human lung cancer cells. (**e**) Meanwhile, culture with 0.5–10 µM of colicin N for 24 h caused no alteration of %cell viability in human dermal papilla DPCs cells. Mode of cell death detected via co-staining of Hoechst33342/propidium iodide (PI) presents apoptosis in lung cancer (**b**) H292 and (**d**) H23 cells treated with 10–15 µM colicin N. (**f**) Neither apoptosis nor necrosis was observed in DPCs cells incubated with colicin N (10–15 µM). Values are means of the independent triplicate experiments ± SD. * *p* ≤ 0.05 versus non-treated control.

**Figure 6 molecules-25-00816-f006:**
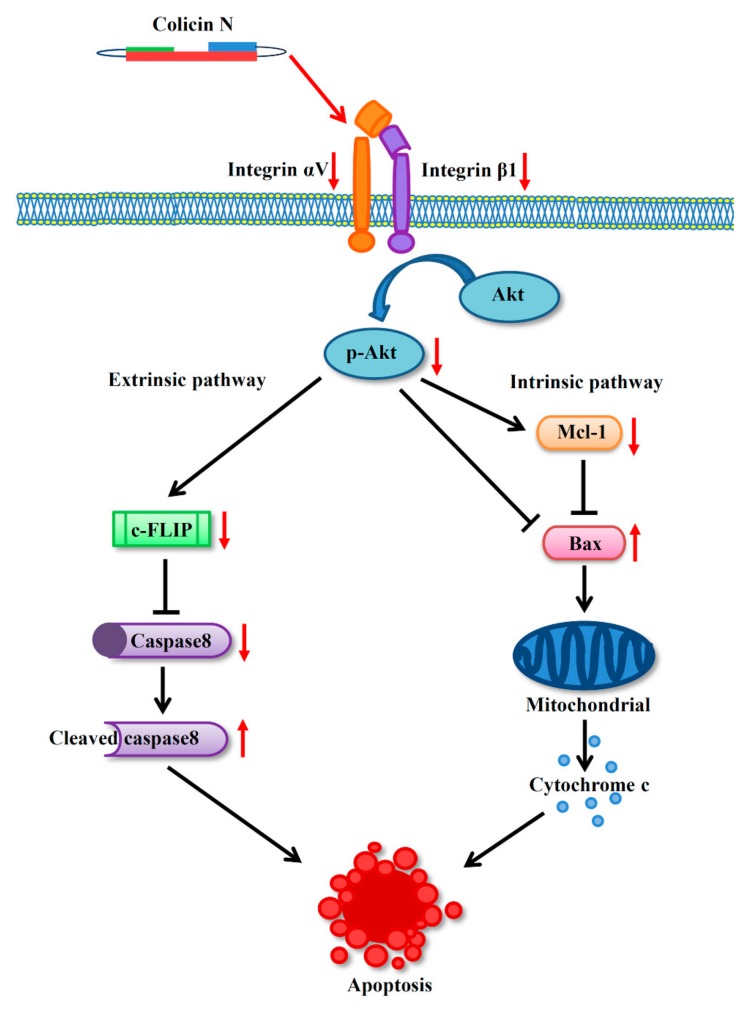
Considering its effect on down-regulation of integrin β1 and αV, colicin N modulates integrin signaling necessary for cell survival evidenced with the decrease in p-Akt, which is the active form of the master survival regulator, Akt. This event shifts the cells towards activation of extrinsic and intrinsic apoptosis by alteration of c-FLIP and the Bcl-2 family proteins (Mcl-1 and Bax), respectively.

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
