# Peer review of "Colicin N Mediates Apoptosis and Suppresses Integrin-Modulated Survival in Human Lung Cancer Cells"

_molecules, 2020, doi:10.3390/molecules25040816_

Round 1
Reviewer 1 Report
RE: Arunmanee et al. titled “Colicin N mediates apoptosis and suppresses integrin 2 modulated-survival in human lung cancer cells”
Main findings
This manuscript described the cytotoxic and mechanistic effects of Colicin N in human lung cancer cell lines. For most parts, the results of the study are well supported by the data presented and the manuscript is well written.
Revisions required for publication
Comments and suggestions:
It is difficult to identify differences in the Hoechst staining, especially Figure 2C. These signals may be just coming from mitotic cells. To confidently conclude DNA condensation/nuclei fragmentation, visualisation in agarose gel electrophoresis using appropriate DNA dye is necessary. It is unclear why the authors have chosen integrin β1 and αV. Are these integrins specifically over-expressed in lung cancer cells? It is also equally unclear whether Colicin N influences other integrins.
Author Response
Response to Reviewers
Reviewer 1
This manuscript described the cytotoxic and mechanistic effects of Colicin N in human lung cancer cell lines. For most parts, the results of the study are well supported by the data presented and the manuscript is well written.
Revisions required for publication
Comments and suggestions:
It is difficult to identify differences in the Hoechst staining, especially Figure 2C. These signals may be just coming from mitotic cells. To confidently conclude DNA condensation/nuclei fragmentation, visualisation in agarose gel electrophoresis using appropriate DNA dye is necessary.Response: The valuable comments and suggestions from reviewer are appreciated. To demonstrate apoptosis cell death detected by nuclear staining assay, the replacement Hoechst33342 staining presenting clearer DNA condensation/nuclei fragmentation and apoptosis body was indicated in the Figure 2c of revised manuscript.
Although nuclear staining with Hoechst33342 and propidium iodide (PI) is wildly accepted for screening mode of cell death, we agree with the reviewer that it is difficult to differentiate apoptosis and mitotic cell via co-staining of Hoechst33342/PI. Therefore, flow cytometry analysis of annexin V-FITC/PI, a more specific method for detection of mode of cell death was further performed to confirm apoptosis in this study as presented in Figure 3.
It is unclear why the authors have chosen integrin β1 and αV. Are these integrins specifically over-expressed in lung cancer cells? It is also equally unclear whether Colicin N influences other integrins.Response: According to the objective of investigation on cytotoxicity and apoptosis-inducing effects as well as related anticancer mechanisms of colicin N in human non-small cell lung cancer cells, the rationale to evaluate integrin β1 and αV was added in the Introduction section of revised manuscript as “The overexpression of integrin β1 has been shown to be a poor prognostic factor in NSCLC associated with drug resistance [20, 31]. β1 integrin is particularly important in lung cancer as several studies suggest, and one of its prominent binding partners is αV integrin which has also been noted to be up-regulated in lung cancer [48]. The overexpression of αVβ1 has also been implicated in the resistance of lung cancer cells against radiation and various chemotherapeutic agents [31]. Taken together, the modulation of integrin β1 and αV is a worthy strategy in the search for novel and improved therapeutic options for lung cancer treatment”
Due to our focus on survival and drug resistance in human lung cancer cells, the possible mechanism of colicin N on other sub-types of integrin were excluded in this study. Nevertheless, the potential effects of colicin N on aggressive features mediated by integrins were proposed in the Discussion section of revised manuscript as “Among many sub-types of integrins, β1 integrin in coordination with Akt is notoriously exploited by NSCLC to acquire resistance to cell death induced by anticancer drugs [20, 50]. Overexpression of integrin αV correlates with metastasis lesion in various cancer cells including lung cancer [51]. It is worth noting that the modulation on other sub-types of integrin and extended benefits on metastasis prevention and chemo-sensitization of colicin N are interested in future studies.”
Typing and format correction
The minor change of typing error and manuscript format was corrected in the revised manuscript as following:
Author affiliation of E.P. was corrected and added. Spelling in figure 6 was corrected as presented in revised_figure6 and revised manuscript. Format of reference list was changed according to MDPI format style. English grammar and spelling in the manuscript was corrected and proofed by English native speaker.Reviewer 2 Report
The work of W. Arunmanee et al “Colicin N mediates apoptosis and suppresses integrin 3 modulated-survival in human lung cancer cells” had shown very interesting cytotoxic effect on malignant human cell lines of bacteria-produced Colicin N, the peptide antibiotic with known pore-forming type of toxicity. Not only colicins exert cytotoxicity on tumor cells, with the pore-formers showing the strongest effect, but many other bacterial pore-formers and even natural innate pore formers can do this (Chumchalova, Smarda, 2003; Lagos et al, 2003, Kucheryavykh et al, 2019). Some of pore former peptides disguise itself as essential “food” elements, for example, microcin E492 is recognized by catecholate siderophore receptors as such and translocated across the outer membrane to exert “killing”action (Lagos et al, 2003). The presented work is very interesting and must be published, because it shows clear effect of colicin N selectively causing cytotoxicity in human lung cancer cells lines with no noticeable cell death in human dermal papilla cells. These are the important practical results. While authors has shown that Colicin N exhibits selective anticancer activity associating with suppression on integrin-modulated survival, it explain only the effects on drug resistance. This reviewer suggest that at list some additional discussion paragraph must be added to the article explaining how pore-former may exert cytotoxicity mainly in cancer cells, that in case of Colicin N probably need additional study.
This reviewer also suggest that new natural peptide without side effects and without known resistance can be effective anticancer agents, and the presented work is important careful study and will be interestining to many researchers in the field.
The work is well written in good English. This reviewer found only one phrase that I suggest to change to make it more clear: Line 261-262
None of apoptosis found in human dermal papilla DPCs cells treated with colicin N strongly supports the development of colicin N as an selective anticancer drug for treatment of lung cancer.
This reviewer suggest to change the word ”none” to “absence”.
Author Response
Response to Reviewers
Reviewer 2
The work of W. Arunmanee et al “Colicin N mediates apoptosis and suppresses integrin modulated-survival in human lung cancer cells” had shown very interesting cytotoxic effect on malignant human cell lines of bacteria-produced Colicin N, the peptide antibiotic with known pore-forming type of toxicity. Not only colicins exert cytotoxicity on tumor cells, with the pore-formers showing the strongest effect, but many other bacterial pore-formers and even natural innate pore formers can do this (Chumchalova, Smarda, 2003; Lagos et al, 2003, Kucheryavykh et al, 2019). Some of pore former peptides disguise itself as essential “food” elements, for example, microcin E492 is recognized by catecholate siderophore receptors as such and translocated across the outer membrane to exert “killing”action (Lagos et al, 2003). The presented work is very interesting and must be published, because it shows clear effect of colicin N selectively causing cytotoxicity in human lung cancer cells lines with no noticeable cell death in human dermal papilla cells. These are the important practical results. While authors has shown that Colicin N exhibits selective anticancer activity associating with suppression on integrin-modulated survival, it explain only the effects on drug resistance.This reviewer suggest that at list some additional discussion paragraph must be added to the article explaining how pore-former may exert cytotoxicity mainly in cancer cells, that in case of Colicin N probably need additional study.
This reviewer also suggest that new natural peptide without side effects and without known resistance can be effective anticancer agents, and the presented work is important careful study and will be interestining to many researchers in the field.
Response: The suggestions of the reviewer have been noted and carried out. An additional paragraph explaining about possible factors affecting selective cytotoxicity of colicin N in human lung cancer cells has been added in the Discussion section of revised manuscript as “Colicin N is a cationic antimicrobial pore-forming protein, and like most colicins, its antimicrobial mechanism has been thoroughly investigated. Interestingly, it has been suggested that pore-forming colicins induce cytotoxicity and apoptosis by generating pores in the plasma membrane of cancer cells [23, 26]. Bacteriocins such as colicins are deemed to exert selectivity towards cancer cells due to surface factors that differentiate cancer cells from normal cells [23]. Consideration on negative charge and overexpression of specific integrins on cell membrane of lung cancer cells is a point of interest that should be further investigated for selective anticancer activity of colicin N.”
The work is well written in good English. This reviewer found only one phrase that I suggest to change to make it more clear: Line 261-262None of apoptosis found in human dermal papilla DPCs cells treated with colicin N strongly supports the development of colicin N as an selective anticancer drug for treatment of lung cancer.
This reviewer suggest to change the word ”none” to “absence”.
Response: As suggested, the sentence pointed out by the reviewer has been altered to “The absence of apoptosis in human dermal papilla DPCs cells treated with colicin N strongly supports the development of colicin N as a selective anticancer drug for treatment of lung cancer.”
Typing and format correction
The minor change of typing error and manuscript format was corrected in the revised manuscript as following:
Author affiliation of E.P. was corrected and added. Spelling in figure 6 was corrected as presented in revised_figure6 and revised manuscript. Format of reference list was changed according to MDPI format style. English grammar and spelling in the manuscript was corrected and proofed by English native speaker.Reviewer 3 Report
The authors report herein a study on the potential application of the antibiotic peptide Colicin N for cancer treatment. The study is well conducted and the explanation is clear.
The principal issue concerns the demonstation of colicin interaction with integrin receptors. In Figure 4e, the down-regulation of two classes of integrins is reported and, lately at page 8 lines 246-255, a possible explanation of the mechanism is suggested. The authors hypothesize that disruption of integrin-ECM proteins' interaction by colicin (as antagonist) could induce internalization and receptor intracellular degradation. The topic of integrin trafficking has been object of great debate in the scientific community, anyway the effective binding of colicin to integrin receptors is herein only suggested. In my opinion the authors should perform cell adhesion inhibition experiments in the presence of fibronectin or vitronectin ECM proteins, and verify if the presence of selective antibodies to theì integrin receptors inhibits the observed phenomena on Akt and following cascade.
Some minor suggestions:
page 2, line 80 some more details on Colicin should be added page 4 line 147 and page 5 line 160 , an explanation on the weak behavior of Bcl-2 and Erk respectively should be added all the figures are shifted forward in respect to the text. I wouls suggest to anticipate them all. Figure 6 should be moved before the conclusion section, where a longer and more detailed summary should be given.In my opinion, after these changes and additions, the paper will be suitable for pubblication.
Author Response
Response to Reviewers
Reviewer 3
The authors report herein a study on the potential application of the antibiotic peptide Colicin N for cancer treatment. The study is well conducted and the explanation is clear.The principal issue concerns the demonstation of colicin interaction with integrin receptors. In Figure 4e, the down-regulation of two classes of integrins is reported and, lately at page 8 lines 246-255, a possible explanation of the mechanism is suggested. The authors hypothesize that disruption of integrin-ECM proteins' interaction by colicin (as antagonist) could induce internalization and receptor intracellular degradation. The topic of integrin trafficking has been object of great debate in the scientific community, anyway the effective binding of colicin to integrin receptors is herein only suggested. In my opinion the authors should perform cell adhesion inhibition experiments in the presence of fibronectin or vitronectin ECM proteins, and verify if the presence of selective antibodies to theì integrin receptors inhibits the observed phenomena on Akt and following cascade.
Response: The valuable comments and suggestions from the reviewer are very appreciated. We agree that the topic of integrin trafficking has been object of great debate in the scientific community, anyway the effective binding of colicin to integrin receptors is herein only suggested. Although the inhibitory effect of colicin N on cell adhesion was not directly evaluated in this study, rounded morphology of lung cancer cells treated with colicin N for 24 h was observed (data not shown). These were emphasized in the Discussion section of revised manuscript as “Antagonistic effect on specific integrin motifs has been proposed to disrupt integrin-ECM interaction following with internalization and eventually degradation of integrins [48, 49]. Despite observed round morphology of colicin N treated-human lung cancer cells (data not shown), the inhibitory effect of colicin N on the interaction between specific integrins and ECM components consequence with the alteration of downstream signaling molecules need to be further investigated.”
Some minor suggestions:2.1) page 2, line 80 some more details on Colicin should be added, page 4 line 147 and page 5 line 160 , an explanation on the weak behavior of Bcl-2 and Erk respectively should be added.
Response: The information about anticancer study of colicin was added in the Introduction section of revised manuscript as “Although, accumulating research in colicins are warranted, as in vitro studies revealed apoptosis induction in several cancer cells and in vivo mice studies have shown reduced tumor burden after colicin treatment, the certain anticancer mechanism of colicins is not well established [23, 26].”
The explanation about weak behavior of Bcl-2 was added under sub-topic of Colicin N activates both intrinsic and extrinsic apoptosis pathways in the Results section of revised manuscript as “The level of anti-apoptosis protein, Mcl-1, obviously decreased in H460 cells after incubation with 10-15 µM colicin N while there was no alteration of Bcl-2 protein level (Figure 4c). Mitochondrial permeabilization during apoptosis is dependently on Bax oligomerization which can be regulated by Mcl-1 [44]. Correlation with reduction of Mcl-1, figure 4d indicates the up-regulation of Bax in H460 cells exposed with 10-15 µM colicin N for 12 h. The obtained results imply that colicin N mediates intrinsic apoptosis via modulation of Mcl-1 and Bax expression level.”
The explanation about weak behavior of Erk was added under sub-topic of Inhibitory effect of colicin N on integrin/Akt survival signal in human lung cancer cells in the Results section of revised manuscript as “Moreover, colicin N at 5-15 µM did not remarkably change the expression of Erk and p-Erk in human lung cancer cells. Considering this, the decrease in cell viability and survival in colicin N treated-lung cancer cells might be primarily mediated by integrin/Akt signaling pathway.”
2.2) all the figures are shifted forward in respect to the text. I wouls suggest to anticipate them all. Figure 6 should be moved before the conclusion section, where a longer and more detailed summary should be given.
Response: All figures in the revised manuscript were reposition as suggestion by reviewer. Moreover, the detailed summary was given in the Conclusion section of the revised manuscript as “The down-regulation of αV and β1 integrins which are frequently overexpressed in lung cancer may also provide the additional benefit of circumventing drug resistance often encountered in current therapy. Additionally, the selective toxicity of colicin N as evidenced by its low toxicity towards actively-dividing human dermal papilla cells indicate its advantage over conventional chemotherapy in terms of safety profile.”
Typing and format correction
The minor change of typing error and manuscript format was corrected in the revised manuscript as following:
Author affiliation of E.P. was corrected and added. Spelling in figure 6 was corrected as presented in revised_figure6 and revised manuscript. Format of reference list was changed according to MDPI format style. English grammar and spelling in the manuscript was corrected and proofed by English native speaker.Round 2
Reviewer 1 Report
The manuscript is now suitable for publication.
Reviewer 3 Report
The revised version can now be accepted for publication